# Numerical Simulation of an Air-Bubble System for Ice Resistance Reduction

**Bao-Yu Ni [1],\*** , **Hongyu Wei [1]**, **Zhiyuan Li [2],\*** , **Bin Fang [3]** and **Yanzhuo Xue [1]**

[1]   College of Shipbuilding Engineering, Harbin Engineering University, Harbin 150001, China
[2]   Department of Mechanics and Maritime Sciences, Chalmers University of Technology,
      SE-41296 Gothenburg, Sweden
[3]   Marine Design & Research Institute of China, Shanghai 200011, China
\*    Correspondence: nibaoyu@hrbeu.edu.cn (B.-Y.N.); zhiyuan@chalmers.se (Z.L.)

**Abstract:** Ships sailing through cold regions frequently encounter floe ice fields. An air-bubble system that reduces friction between the hull and ice floes is thus considered useful for the reduction of ice-induced resistance. In this study, a numerical analysis procedure based on coupled finite volume method (FVM) and discrete element method (DEM) is proposed to simulate complicated hull-water-gas-ice interactions for ice-going ships installed with air-bubble systems. The simulations reveal that after turning on the air-bubble system ice floes in contact with the hull side wall are pushed away from the hull by the gas-water mixture, resulting in an ice-free zone close to the side hull. It is found that the drag reduction rate increases with the increase of ventilation, while the bow ventilation plays a deciding role in the overall ice-resistance reduction. The proposed procedure is expected to facilitate design of new generations of ice-going ships.

**Keywords:** air-bubble system; floe ice field; ice resistance; numerical simulation; coupled CFD-DEM

## 1. Introduction

As a consequence of global warming, the polar marginal ice zone has been observed to become wider [1]. The polar marginal ice zone is characterized by floe ice fields that cover 15–80% sea surface [2]. The existence of ice in water induces extra resistance for ships sailing in floe ice fields [3–5]. During the hull-ice interaction, significant kinetic energy from ship propulsion is dissipated, resulting in speed loss or additional fuel consumption [6,7]. An air-bubble system that reduces the friction between the hull and the ice floes is thus considered useful for the reduction of ice-induced resistance in floe ice fields.

The air-bubble system discussed in this article has something in common with but differs from the air lubrication technology that has been utilized for drag reduction. The ship's air lubrication technology can be subdivided into two main categories. The first group is termed bubble-induced skin-friction drag reduction (BDR), for which a large number of micro-bubbles are injected into the boundary layer. The second group is termed air layer drag reduction (ALDR), which forms a continuous gas layer on the hull surface [8]. Both groups utilize air as a lubricant, which has been proven to decrease the friction between the ship and the seawater, see, e.g., in [9,10]. The air-bubble system of this study is more similar to the former group of air lubrication technology, but aims to reduce hull-ice friction instead. This is achieved by injecting air from a series of nozzles at the bow and bilge. When the air bubbles arise along the hull, the mixed air and water create a strong current, forming a layer between the hull and the ice floes, consequently reducing ice resistance.

The idea of reducing ice resistance by air bubbles originated in the late 1960s [11]. An air-bubble system was studied mainly through model tests. Till the early 1990s, several icebreakers and ice-going vessels that mainly operate in the Baltic Sea were installed with air-bubble systems [12]. Little progress in air-bubble systems for ice resistance reduction

has been reported in recent decades. Nevertheless, as more ships sail into the polar floe ice fields, an air-bubble system may bring added value to ice-going ships' efficiency and safety, and thus deserves further investigation. Furthermore, the fast development of computational fluid dynamic (CFD) methods makes it possible to investigate complicated hull-water-air-ice interaction processes with numerical simulations, instead of depending on high-cost model tests. In this work, the authors aim to make use the state-of-the-art numerical methods to quantify the ice resistance characteristics of ice-going ships installed with air-bubble systems.

There are two major numerical methods for simulating air-bubbles. The first one is termed the interface tracking method [13]. With this approach, the fluid interface can be accurately defined. However, this method requires a complicated process of mesh reconstruction, and it has the disadvantage of mass and energy loss of the bubbles. The second method is called the interface capturing method [14], in which the fluid interface does not have to be accurately defined. The different liquids are instead distinguished through additional fluid variables such as the mass fraction. The interface capturing method requires a large number of grid cells to keep the accuracy. This method is represented by the method of the volume of fluid (VOF) approach, which is often employed in the simulation of large bubble motions and free surface flow in liquids [15]. Some recent research making use of the VOF approach are as follows: Zhu et al. [16] used the VOF method to investigate the effects of gas velocity, liquid velocity and other factors on the bubble detachment diameter. Based on the VOF method, Li et al. [17] described the deformation during the ascent of a single bubble in gas-liquid, gas-liquid-solid multiphase flow under high pressure. Tsui et al. combined the VOF method with the Level Set method to simulate rising bubbles in still water and got agreeable results [18].

Numerical simulations of ship resistance characteristics in ice-infested waters have been presented by many researchers. A recent review paper by Li and Huang [19] indicates that the discrete element method (DEM) predicates reasonable resistance induced by broken ice. Hansen and Loset [20] applied a two-dimensional DEM model to simulate the ice force on ships in broken ice. Ji et al. [21,22] used the GPU parallel algorithm to accelerate the DEM calculation, making it possible to use the DEM method to calculate sea ice structure interaction in the large-scale calculation domain. Luo et al. [23] applied combined CFD-DEM to study the coupling characteristics of ship-ice-water in the brash ice channel and analyzed the difference between one-way coupling and two-way coupling. In addition to the DEM approach, other numerical methods were also employed for ship-ice interaction simulations. Kim et al. [24] simulated ice resistance in ice channels through the finite element method (FEM) and found the simulation results were in good agreement with the model test results. Lubbad and Loset [25] simulated ships in ice through the physics engine PhysX, and compared it with the full-scale measurements. Furthermore, there are other emerging numerical methodologies, such as the Peridynamics (PD) method, the Smooth Particle Hydrodynamics (SPH) method, and the Extended Finite Element (XFEM) method. All those numerical methods have the potential to simulate ship-ice interactions with reasonable accuracy [19,26].

In this paper, the authors utilized the combined CFD-DEM approach to simulate ship resistance in floe ice fields with the air-bubble system installed. The air-water interface was simulated by using the VOF method. An icebreaker was chosen as the case study vessel. By this means, we aim to find out how effective the air bubbles are for drag reduction in ice-infested waters.

## 2. The Numerical Models

In this section, the features of the numerical models of this study are summarized and the key theoretical formulations are presented as follows.

### 2.1. The Governing Equations of the Incompressible Fluid

In this study, the finite volume method (FVM) is used to discretize the fluid domain. The governing equations are:

$$\frac{\partial}{\partial t} \int_V \rho dV + \oint_A \rho \mathbf{v} \cdot d\mathbf{a} = \int_V S_u dV \tag{1}$$

$$\frac{\partial}{\partial t} \int_V \rho \mathbf{v} dV + \oint_A \rho \mathbf{v} \otimes \mathbf{v} \cdot d\mathbf{a} = -\oint_A p\mathbf{I} \cdot d\mathbf{a} + \oint_A \mathbf{T} \cdot d\mathbf{a} + \int_V \mathbf{f}_b dV + \int_V \mathbf{s}_u dV \tag{2}$$

where $t$ is time, $V$ is the volume of the fluid element, $\mathbf{a}$ is the area vector, $\mathbf{v}$ is the velocity vector of the fluid element, $S_u$ is the source term of the continuity equation, $p$ is the pressure, $\mathbf{T}$ is the viscous stress tensor, $\mathbf{f}_b$ is the resultant force of the body force, $\mathbf{s}_u$ is the source term of the momentum conservation equation. The viscous stress tensor can be expressed as:

$$\mathbf{T} = \mu(\nabla \mathbf{v} + (\nabla \mathbf{v})^T) - \frac{2}{3}\mu(\nabla \cdot \mathbf{v})\mathbf{I} \overset{\text{incompressible flow}}{\longrightarrow} \mathbf{T} = \mu(\nabla \mathbf{v} + (\nabla \mathbf{v})^T) \tag{3}$$

where $\mu$ is the dynamic viscosity coefficient, $\mathbf{I}$ is the unit tensor.

### 2.2. The Governing Equations of the Discrete Phase in Numerical Simulation

The governing equations of the discrete phase follow the Lagrangian framework. The surface force and physical force acting on the particle jointly determine the change of particle momentum, and its momentum conservation equation is:

$$m_p \frac{d\mathbf{v}_p}{dt} = \mathbf{F}_s + \mathbf{F}_b = \mathbf{F}_d + \mathbf{F}_p + \mathbf{F}_{vm} + \mathbf{F}_g + \mathbf{F}_{con} \tag{4}$$

In this equation, $m_p$ is particle mass, $\mathbf{v}_p$ is particle velocity, $\mathbf{F}_s$ is surface force, $\mathbf{F}_b$ is body force, $\mathbf{F}_d$ is the drag force, $\mathbf{F}_p$ is pressure gradient force, $\mathbf{F}_{vm}$ is virtual mass force, $\mathbf{F}_g$ is the gravity, $\mathbf{F}_{con}$ is the contact force. The two most critical items are the calculation of the drag force and the contact force. The former involves the treatment of the gas-liquid mixed phase, and the latter depends on the choice of the contact model. The calculation of both will be introduced in subsequent sub-sections.

The conservation of angular momentum of the particle can be expressed as:

$$\mathbf{I}_p \frac{d\omega_p}{dt} = \mathbf{M}_b + \sum_i (\mathbf{r}_c \times \mathbf{F}_{ci} + \mathbf{M}_{ci}) \tag{5}$$

where $\mathbf{I}_p$ is the moment of inertia of the particle, $\omega_p$ is the angular velocity of the particle, $\mathbf{M}_b$ is the resistance moment, $\mathbf{r}_c$ is the vector from the contact point to the center of gravity, $\mathbf{F}_{ci}$ is the contact force between particle c and particle $i$, and $\mathbf{M}_{ci}$ is the moment of rolling resistance acting on the particle.

### 2.3. The Turbulence Model and the Free Surface Treatment

The turbulence model of the RANS equation used in the numerical simulation of this paper is the standard $k$-$\varepsilon$ model. The existence of the turbulence model is to make the Reynolds-averaged Navier-Stokes equation closed. For the RANS equation, the average value can be regarded as the time average of the steady-state situation and the overall average of repeatable transient situations. Inserting the decomposed solution variables into the Navier-Stokes equations yields equations of average quantities. The conservation equations of average mass and average momentum can be expressed as:

$$\frac{\partial \rho}{\partial t} + \nabla \cdot (\rho \bar{v}) = 0 \tag{6}$$

$$\frac{\partial}{\partial t}(\rho\overline{v}) + \nabla \cdot (\rho\overline{v} \otimes \overline{v}) = -\nabla \cdot \overline{p}\mathbf{I} + \nabla \cdot (\overline{\mathbf{T}} + \mathbf{T_{RANS}}) + \mathbf{f_b} \tag{7}$$

where $\rho$ is the density, $\overline{v}$ is the mean velocity, $\overline{p}$ is the mean pressure, $\overline{\mathbf{T}}$ is the mean viscous stress tensor, and $\mathbf{f}_b$ is the resultant force of body forces (such as gravity, centrifugal force, etc.). This equation is essentially the same as the original N-S equation, with an extra term $\mathbf{T_{RANS}}$ added to the momentum equation, which is the stress tensor.

The standard $k$-$\varepsilon$ model is a two-equation model that determines the turbulent length and time scale by solving two independent transport equations. This model assumes a fully turbulent fluid flow and does not take into account the effects of molecular viscosity. The transport equation corresponding to the turbulent kinetic energy and dissipation rate of the standard $k$-$\varepsilon$ model is of the form:

$$\frac{\partial}{\partial t}(\rho k) + \nabla \cdot (\rho k \overline{v}) = \nabla \cdot \left[\left(\mu + \frac{\mu_i}{\sigma_k}\right)\nabla k\right] + G_k + G_b - \rho\varepsilon - Y_M + S_k \tag{8}$$

$$\frac{\partial}{\partial t}(\rho\varepsilon) + \nabla \cdot (\rho\varepsilon\overline{v}) = \nabla \cdot \left[\left(\mu + \frac{\mu_i}{\sigma_\varepsilon}\right)\nabla\varepsilon\right] + C_{1\varepsilon}\frac{\varepsilon}{k}(G_k + C_{3\varepsilon}G_b) - C_{2\varepsilon}\rho\frac{\varepsilon^2}{k} + S_\varepsilon \tag{9}$$

where $G_k$ is the term from the turbulent kinetic energy, $k$ due to the average velocity gradient, $G_b$ is the term from the turbulent kinetic energy caused by the buoyancy effect, $C_{1\varepsilon}$, $C_{2\varepsilon}$ and $C_{3\varepsilon}$ are empirical constants. $\sigma_k$ and $\sigma_\varepsilon$ are the Prandtl numbers corresponding to the turbulent kinetic energy and dissipation rate, respectively. $S_k$ and $S_\varepsilon$ are user-defined source terms.

The relationship between the turbulent kinetic viscosity and turbulent kinetic energy and the dissipation rate can be expressed as:

$$\mu_i = \rho C_\mu \frac{k^2}{\varepsilon} \tag{10}$$

In this equation, $C_\mu$ is the empirical constant. The standard $k$-$\varepsilon$ model is a semi-empirical formula derived from physical experiments combined with theory.

In order to capture the water-air interface better to simulate the effect of the bubble assist system, this paper adopts the VOF (Volume of Fluids) method and HRIC (High-Resolution Interface Capture) format. The VOF method is used to capture incompatible terms and assumes that the mesh resolution is sufficient to resolve the position and shape of the interface between the different phases. Therefore, we should pay attention to the mesh size during the numerical simulation. Figure 1 shows the unsuitable mesh and the suitable mesh:

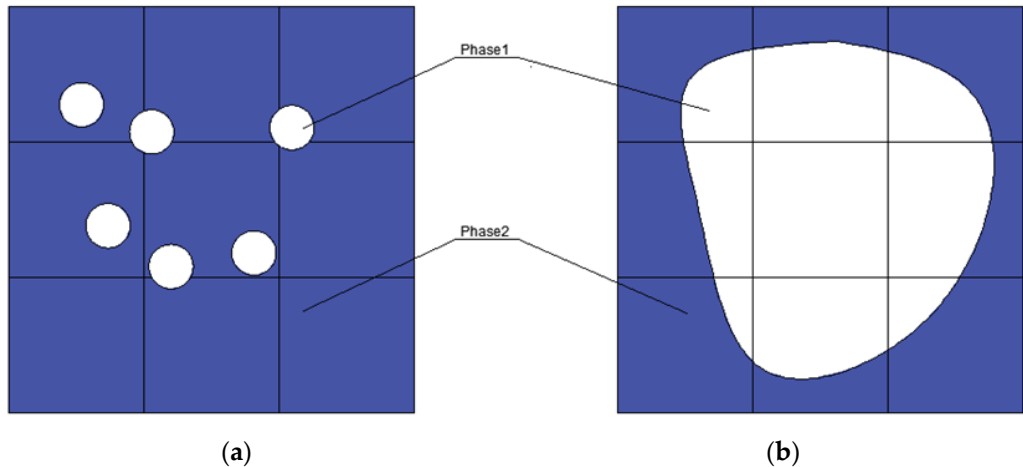

(a)             (b)

**Figure 1.** (**a**) unsuitable mesh size; (**b**) suitable mesh size.

The VOF model describes the phase distribution and position of the interface through the field of the phase volume fraction $\alpha_i$, the volume fraction $\alpha_i = \frac{V_i}{V}$ of the phase $i$, where $V_i$ is the volume of phase $i$ in the grid cell, $V$ is the volume of the grid cell, And the sum of the volume fractions of all phases within each grid cell is 1. When the grid contains only a single fluid, the material properties of the fluid are used in the calculation. If there are multiple fluid phases in the grid, it is regarded as a mixture, and its material properties use the weighted average of each phase.

The distribution of fluid phase $i$ is determined by the mass conservation equation:

$$\frac{\partial}{\partial t}\int_V \alpha_i dV + \oint_A \alpha_i \mathbf{v} \cdot d\mathbf{a} = \int_V \left(S_{\alpha_i} - \frac{\alpha_i}{\rho_i}\frac{D\rho_i}{Dt}\right)dV - \int_V \frac{1}{\rho_i}\nabla \cdot (\alpha_i \rho_i \mathbf{v}_{d,i})dV \tag{11}$$

where $\mathbf{a}$ is the surface area vector, $\mathbf{v}$ is the velocity of the mixed fluid, $\mathbf{v}_{d,i}$ is the diffusion velocity, $S_{\alpha i}$ is the source term of the phase $i$, and $D\rho_i/Dt$ is the Lagrangian derivative of the phase density $\rho_i$. When only two phases, water and air, are present in the simulation, the mass conservation equation is solved for the first term only, and the volume fraction of the second phase is adjusted in each grid cell so that the sum of the volume fractions equals 1.

The momentum equation of the fluid can be expressed as:

$$\frac{\partial}{\partial t}\left(\int_V \rho \mathbf{v} dV\right) + \oint_A \rho \mathbf{v} \otimes \mathbf{v} \cdot d\mathbf{a} = \\ \oint_A (p\mathbf{T} - p\mathbf{I})\cdot d\mathbf{a} + \int_V \rho g dV + \int_V \mathbf{f}_b dV - \sum_i \int_V \alpha_i \rho_i \mathbf{v}_{d,i} \otimes \mathbf{v}_{d,i} \cdot d\mathbf{a} + \int_V S_i^\alpha dV \tag{12}$$

where $p$ is the pressure, $\mathbf{I}$ is the unit tensor, $\mathbf{T}$ is the stress tensor, $\mathbf{f}_b$ is the vector of the body force, $S_i^\alpha$ is the momentum source term of the phase, and $g$ is the gravity acceleration.

The drag force $\mathbf{F}_d$ provided by the mixed fluid can be calculated as:

$$\mathbf{F}_d = 0.5C_d \rho A_p |\mathbf{v}_s| \mathbf{v}_s \tag{13}$$

where $C_d$ is the drag coefficient, $\rho$ is the density of the continuous phase (mixing density for multiphase flow), $\mathbf{v}_s = \mathbf{v} - \mathbf{v}_p$, $\mathbf{v}$ is the instantaneous velocity of the continuous phase, $\mathbf{v}_p$ is the particle slip velocity, and $A_p$ is the projected area of the particle. The drag coefficient $C_d$ in this equation is determined by the Schiller-Naumann correlation, which applies to fluids with bubbles, which is set as:

$$C_d = \begin{cases} \frac{24}{\mathrm{Re}_p}(1 + 0.15\mathrm{Re}_p^{0.687}), \mathrm{Re}_p \leq 1000 \\ 0.44, \mathrm{Re}_p > 1000 \end{cases} \tag{14}$$

where $\mathrm{Re}_p$ is the particle Reynolds number, which is defined as $\mathrm{Re}_p \equiv \frac{\rho |\mathbf{v}_s| D_p}{\mu}$, where $D_p$ is the particle equivalent diameter and $\mu$ is the kinematic viscosity.

### 2.4. The Ice Model

Ice is modeled by using DEM, and the governing equation is Newton's law which has been described in detail in Section 2.2. The contact force in the governing equation needs to be calculated by the contact model. The contact model of DEM will be introduced below.

We employ the DEM method in this study to model the ice floes. Two major assumptions are taken to simplify the computation. Firstly, we assume that the ice floes will be pushed away but not broken during the ship-ice interaction process. Secondly, the contacts of ship-ice and ice-ice are assumed as elastic. Based on these assumptions, the Hertz-Mindlin model can be implemented. In this model, the spring simulates the elastic part of the collision process, and the damper reflects the energy dissipation of the collision process. The contact force between two DEM particles is described by the following equations:

$$\mathbf{F}_{con} = \mathbf{F}_n + \mathbf{F}_t \tag{15}$$

where $\mathbf{F}_{con}$ is the contact force, $\mathbf{F}_n$ is the normal force, $\mathbf{F}_t$ is the tangential force. The normal force is expressed as:

$$\mathbf{F}_n = -K_n d_n - N_n v_n \tag{16}$$

where $K_n$ is the normal spring stiffness, $d_n$ is the overlap of the local normal directions of the contact between the two particles, $N_n$ is the Normal damping, $v_n$ is the normal velocity of the particle.

The normal spring stiffness is:

$$K_n = \frac{4}{3} E_{eq} \sqrt{d_n R_{eq}} \tag{17}$$

The normal damping is:

$$N_n = \sqrt{5 K_n M_{eq}} N_{ndamp} \tag{18}$$

where $E_{eq}$ is the equivalent Young's modulus, $R_{eq}$ is the equivalent radius, $M_{eq}$ is the equivalent particle mass and $N_{ndamp} = \frac{-\ln(C_{nrest})}{\sqrt{\pi^2 + \ln(C_{nrest})^2}}$ is the normal damping coefficient, where $C_{nrest}$ is the normal restitution coefficient.

The tangential direction is defined by:

$$\mathbf{F}_t = \begin{cases} -K_t d_t - N_t v_t, & |K_t d_t| < |K_n d_n| C_{fs} \\ \frac{|K_n d_n| C_{fs} d_t}{|d_t|}, & |K_t d_t| \geq |K_n d_n| C_{fs} \end{cases} \tag{19}$$

where $K_t$ is the tangential spring stiffness, $N_t$ is the tangential damping, $d_t$ is the overlap of the local tangential directions of the contact between the two particles, $C_{fs}$ is the coefficient of static friction.

The tangential spring stiffness is:

$$K_t = 8 G_{eq} \sqrt{d_n R_{eq}} \tag{20}$$

The tangential damping is:

$$N_t = \sqrt{5 K_t M_{eq}} N_{tdamp} \tag{21}$$

where $N_{tdamp} = \frac{-\ln(C_{trest})}{\sqrt{\pi^2 + \ln(C_{trest})^2}}$ is the tangential damping coefficient, in which $C_{trest}$ is the tangential restitution coefficient.

The equivalent radius is:

$$R_{eq} = \frac{1}{\frac{1}{R_A} + \frac{1}{R_B}} \tag{22}$$

The equivalent particle mass is:

$$M_{eq} = \frac{1}{\frac{1}{M_A} + \frac{1}{M_B}} \tag{23}$$

The equivalent Young's modulus is:

$$E_{eq} = \frac{1}{\frac{1-v_A^2}{E_A} + \frac{1-v_B^2}{E_B}} \tag{24}$$

The equivalent shear modulus is:

$$G_{eq} = \frac{1}{\frac{2(2-v_A)(1+v_A)}{E_A} + \frac{2(2-v_B)(1+v_B)}{E_B}} \tag{25}$$

where $M_A$ and $M_B$ are the masses of spheres A and B; $R_A$ and $R_B$ are the radii of the sphere; $E_A$ and $E_B$ are Young's modulus of the sphere; $\nu_A$ and $\nu_B$ represent Poisson's ratio of A and B, respectively. For collisions between particles and walls, the above formula remains the same but assumes that the wall radius and mass are summed $R_{wall} = \infty$ and $M_{wall} = \infty$, so the equivalent radius decreases to $R_{eq} = R_{partical}$, and the equivalent mass decreases to $M_{wall} = M_{partical}$.

### 3. The Computational Domain Settings

The main ship particulars of the case study vessel are listed in Table 1. Figure 2 illustrates the three-dimensional hull model as well as the location of the nozzles of the air-bubble system. It is noticeable that the nozzles are placed on the bow and along the side instead of in the bottom and the keel areas. This is because conventional air-bubble systems aim at reducing the water resistance of the ship, which requires the air-bubble system to cover the wet surface of the hull as much as possible. For that purpose, the bottom and the keel areas need to be covered by air-bubbles. The air-bubble system in this study, in contrast, is supposed to reduce ice resistance instead. It would be sufficient to use a smaller volume of air from the bow/sides to push the crushed ice away from the hull.

**Table 1.** The main particulars of the case study vessel.

| Parameter | Value |
| --- | --- |
| Length overall (m) | 122.5 |
| Length of waterline (m) | 116.2 |
| Breadth (m) | 22.3 |
| Draft (m) | 7.8 |
| Stem angle (°) | 20 |
| Waterline angle (°) | 40 |

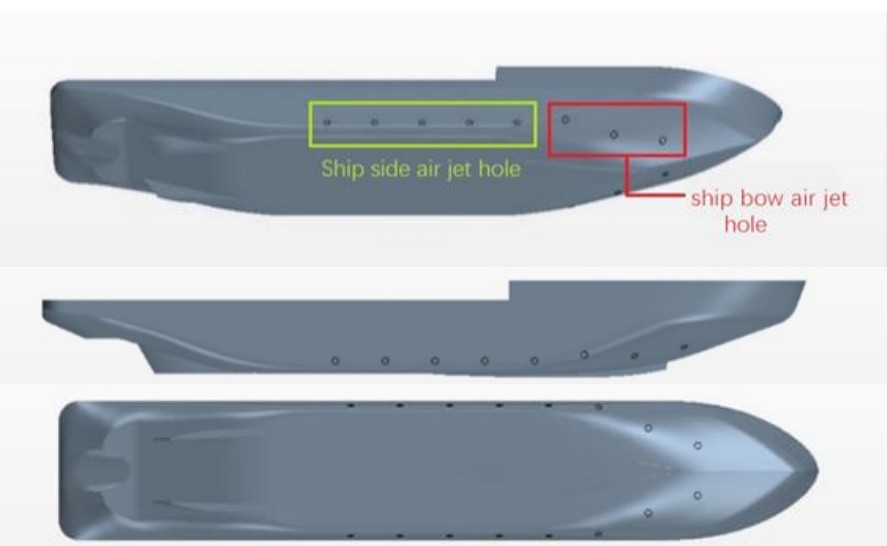

**Figure 2.** The three-dimensional hull model and the nozzle locations of the air-bubble system.

The numerical modelling and simulations are carried out in the commercial CFD software STAR-CCM+. In the calculation domain, the stern bottom is considered the coordinate origin. The positive X-direction is defined as the direction from the stern to the bow; the positive Y-direction is to the port side; the positive Z-direction is upwards. In order to minimize the influence of the boundary on the flow field, the fluid domain should be set as large as possible. In this study, following the experience of a previous study [27], the distances between the boundary from the stern and the bow are set to be twice the ship's length. The water depth is also set to be twice the ship's length. The width of the

ice field is set to be three times the ship's breadth. The ice field is modelled with ice floes made of DEM elements. The size of the ice floe is assumed as 4 m × 4 m × 1 m. The ice concentration is controlled by setting the distance of the injecting point as the injecting speed of the DEM element. The computational domain with a concentration of 60% is illustrated in Figure 3.

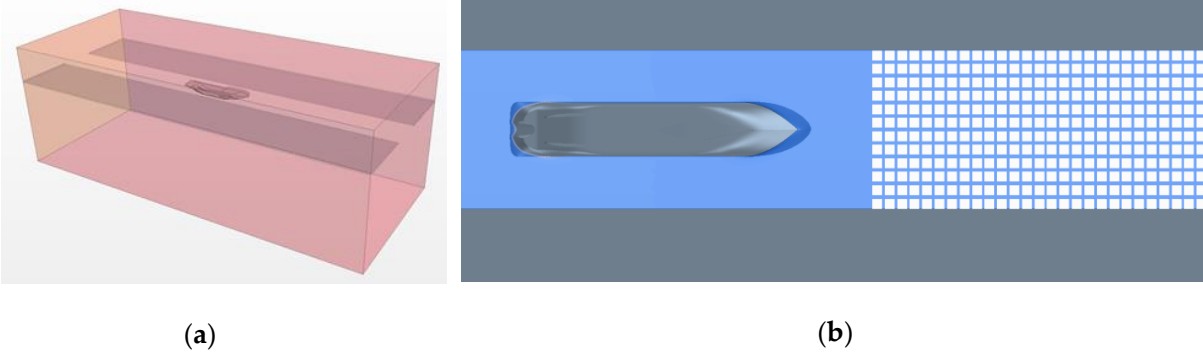

(**a**)  (**b**)

**Figure 3.** Schematic diagram of the computational domain; (**a**) Schematic diagram of the computational domain at the initial stage; (**b**) the layout of the ice field with an ice concentration of 60%.

Table 2 lists the boundary conditions of the computational domain. The boundary to the right is set as the velocity inlet, while the boundary to the right is the outlet. The other surrounding boundary surfaces are set as slip wall conditions. The hull surface and the sides of the ice field are set as non-slip wall conditions. The air inlet is set as the velocity inlet boundary.

**Table 2.** The boundary conditions of the computational domain.

| Boundary | Condition |
|---|---|
| Water inlet | The surface current velocity equals to the ship's speed; the turbulent intensity is 0.01 |
| Water outlet | DEM particle outlet boundary; the outlet pressure is the hydrodynamic pressure; |
| The other boundaries | Slip wall |
| Ship hull | Non-slip wall |
| Air inlet | Air-water interface; the air velocity is the flow rate corresponding to the ventilation volume |
| Ice region | Non-slip wall |
| Free surface | VOF free surface |
| Particle inject | Surface injection conditions in component injectors |

For the computational domain and ship model, the fluid domain meshes with the trimmed meshing model and the boundary layer grids are divided around the ship's surface. The y+ value is in the range of 30–60. The meshes on the ship's surface are refined near the waterline, the stem, the stern, and around the nozzles as shown in Figure 4. The total mesh number is about 3.9 million.

The numerical simulation is carried out by discrete solution of N-S equation based on finite volume method, and the multiphase flow model adopts the VOF method to realize interface tracking [28]. In this paper, there are two kinds of fluids in the computational domain, $\alpha_0$ represents the volume function of the air phase and $\alpha_1$ represents the volume function of the water phase. In the computational domain, the sum of the volume fractions of the two phases is 1 ($\alpha_0 + \alpha_1 = 1$). The turbulence model adopts the standard *k-ε* Model [29].

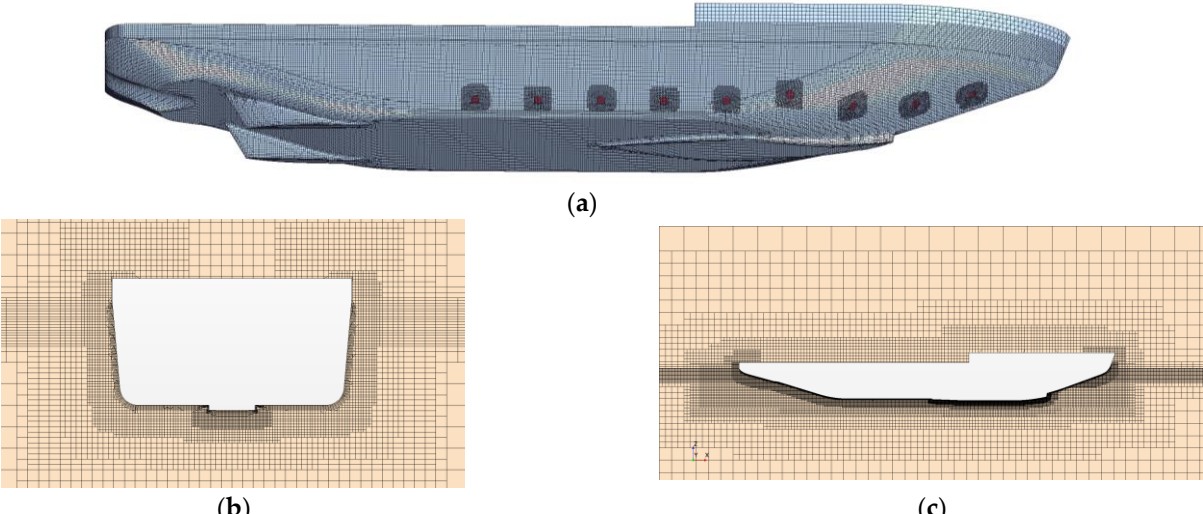

**Figure 4.** The meshing of the numerical model. (**a**) Ship hull mesh; (**b**) Front view of the flow domain; (**c**) Side view of the flow domain.

The nozzle on the hull surface is set as the velocity inlet of the air. In this study, the gas velocity and pressure of the air-bubble system are small that the compressibility is ignored. The separation flow model is employed to describe the liquid phase and gas phase, which solves the momentum equation corresponding to each dimension, and associates the momentum equation with the continuity equation through the prediction correction method. The second-order discretization of the convective flux is in use, which is deemed particularly suitable for constant-density fluids. Each nozzle contains at least 25 complete meshes to avoid excessive numerical loss when the computational domains are generated. Figure 5 illustrates the numerical losses of the computational domains under different mesh numbers. The color in the figure represents the volume fraction of air, red represents the gas volume ratio of 100%, yellow represents the gas-liquid interface (gas volume ratio is 50%), and green color represents the gas attached to the surface of the hull.

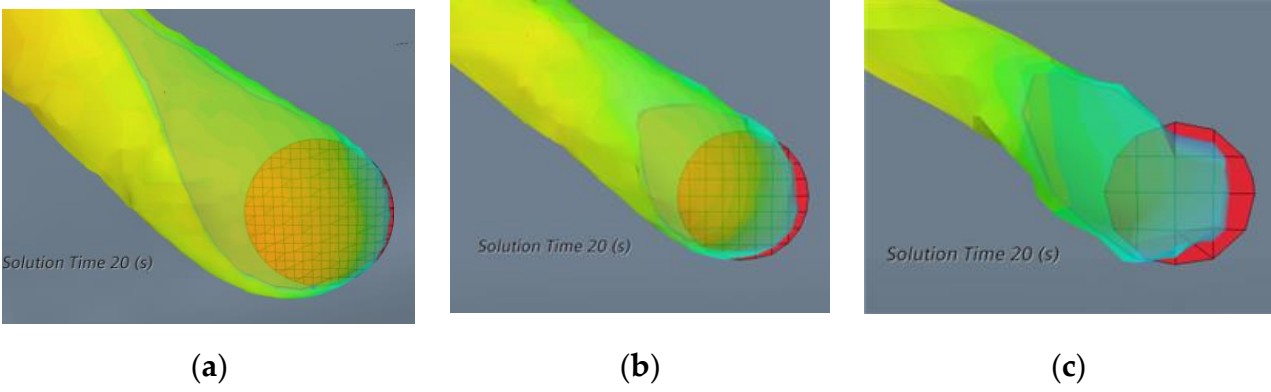

**Figure 5.** The numerical loss of the computational domains under different mesh numbers: (**a**) 100+ complete meshes; (**b**) 25+ complete meshes; (**c**) 4+ complete meshes.

The ice floes are modelled using the DEM element, following the theoretical models as described in Section 2.4. In this work, the ice density is set as 900 kg/m$^3$, Young's modulus is assumed as 1 GPa and Poisson's ratio is assumed as 0.3.

## 4. Simulation Results and Analyses

### 4.1. Comparision of Simulations

Simulations with the air-bubble system activated are compared with those without the air-bubble system under identical operational conditions. For the air-bubble system, the air inlet velocity is 2.5 m/s, and the air jet direction is perpendicular to the hull surface. The ship speed is 6 knots. Figure 6 illustrated the wave patterns and streamlines around the hull for the cases of when the air-bubble system is deactivated and when the air-bubble system is activated, respectively. When the air-bubble system is turned on, the wave pattern around the hull is found to differ obviously from that when the air-bubble system is turned off. After the gas is pumped out from the nozzles, the gas-water mixture rises along the side of the hull and then the gas escapes from the free surface, resulting in a more distorted free surface around the hull. It is also observed that when the gas-water mixture exists the streamlines differ from those when there is no gas in the fluid flow.

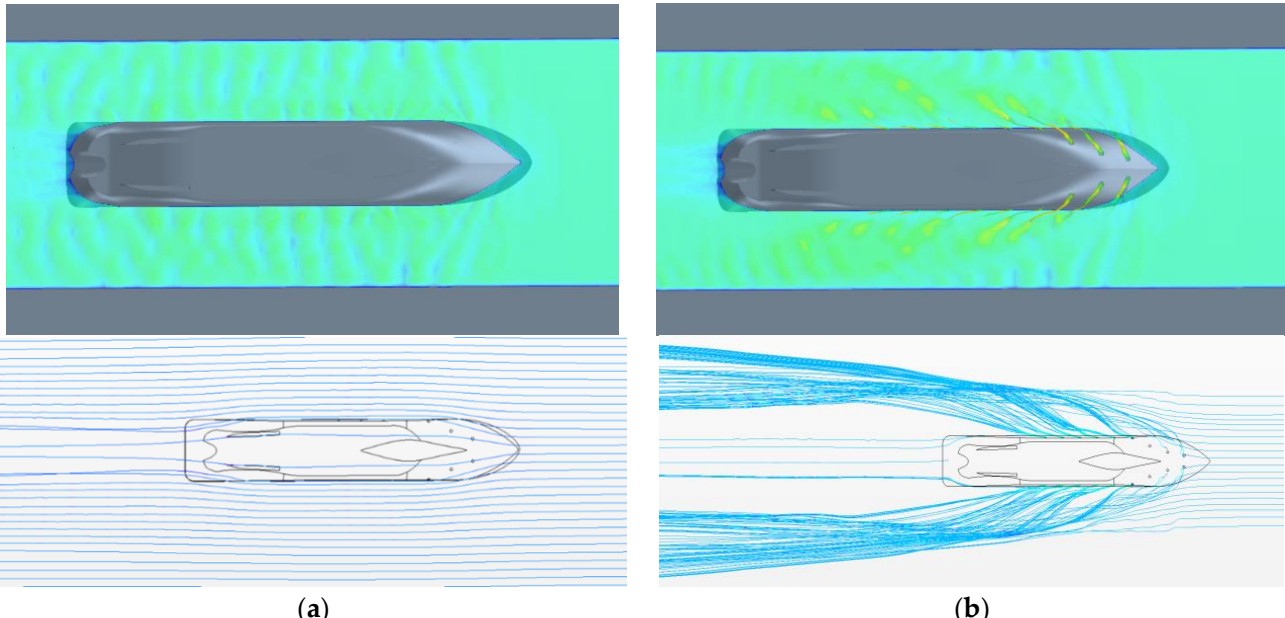

(**a**)　　　　　　　　　　　　　　　　　(**b**)

**Figure 6.** Wave patterns and streamlines around the hull for: (**a**) when the air-bubble system is turned off; (**b**) when the air-bubble system is activated.

When ice exists in the water, additional resistance is induced by the interaction between the hull and the ice floes. In this study, the ice fields are assumed to be composed of ice floes of identical size and shape of 4 m × 4 m × 1 m. The ice concentration is assumed to be 60%. Figure 7 illustrates the snapshot at 120 s of the simulation in the ice field with a ship speed of 6 knots. The air-bubble system has not been activated. When a ship enters the floe ice fields, the speeds of the ice floes around the bow drop rapidly due to wave-making and the collision with the bow. Consequently, the ice floes accumulate around the bow and some of them slide then along the bow area to the ship's sides and the bottom, as shown in Figure 7. It is observed that during the hull-water-ice interaction, the ice floes are overturned by the wave system rising from the bow/shoulder area. Some of the ice floes collide with the bow and also with other ice floes. Then some ice floes move along the ship's side or the bottom, resulting in friction forces on the hull. An ice-free channel slightly narrower than the width of the ship is formed behind the ship.

In contrast, when the air-bubble system is turned on, the hull-water-ice interaction becomes significantly different, which is illustrated in Figure 8. It is observed that when the air-bubble system is on, despite ice accumulation remaining unchanged around the bow, much fewer ice floes become in contact with slide through the shoulder due to the gas-water mixture. Most of the ice floes are overturned before passing through the ship's shoulder

and drifting away from the hull, which greatly reduces the hull-ice contact occurrence at the sides and the bottom.

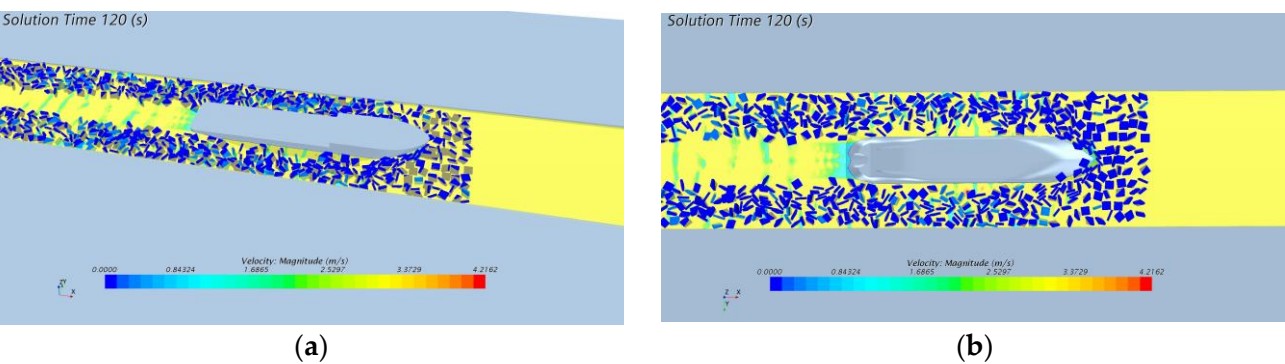

**Figure 7.** The snapshot at 120 s of the simulation when the air-bubble system has not been activated: (**a**) isometric view; (**b**) bottom view.

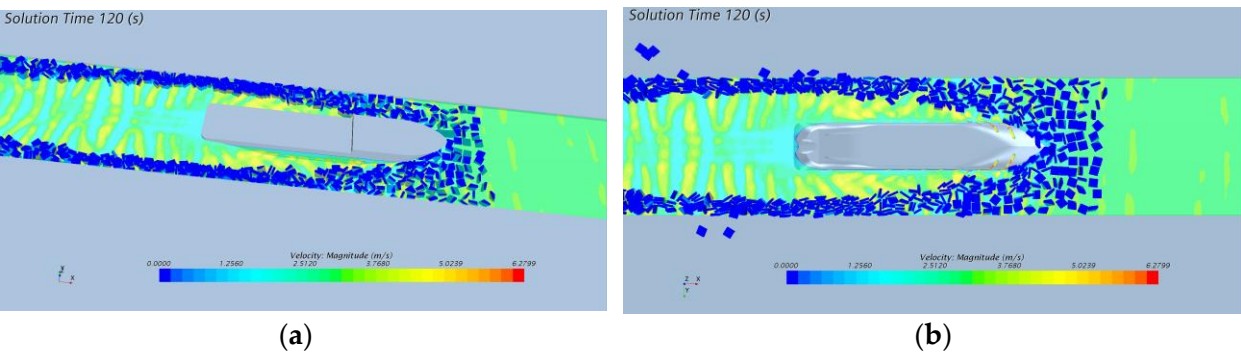

**Figure 8.** The snapshot at 120 s of the simulation when the air-bubble system has been activated: (**a**) isometric view; (**b**) bottom view.

*4.2. Ice Resistance Calculation*

In this subsection, we quantify ice-induced resistance for the cases with and without the air-bubble system. The top subplot of Figure 9 shows the time history of the total ice resistance when the air-bubble system is on, for which t = 11.9 s is the timestep when the ship bow reaches the ice field. It is observed that the ice resistance gradually increases and becomes stable around t = 20 s, which corresponds to the timestep when the entire hull has entered the ice field. The middle and bottom sub-plots of Figure 9, illustrate the time series of the ice resistance components from the bow and the ship's sides, respectively. The ice resistance values of the stable stage, i.e., 20–120 s are listed in Table 3. The bow area accounts for a major part of the total ice resistance.

As mentioned previously, when the air-bubble system is turned on with the air injection rate of 2.5 m/s, the hull-ice contact on the ship side is greatly reduced. Figure 10 illustrates the time series of the ice resistances for this case. In comparison with the ice resistances in Figure 9, the total resistance as well as the components from the bow area and the sides are found to be smaller. The resistance values are listed in Table 3, together with the resistance when the air-bubble system is off. The resistance reductions are also included in Table 3. It is seen that when the air-bubble system is off, the total ice resistance is reduced by 15.3%. When it comes to the ice resistance components, the bow resistance remains almost unchanged, with a reduction rate of 10.3%. In contrast, the ice resistance from the ship's sides is greatly reduced, with a drag reduction rate of 70.8%.

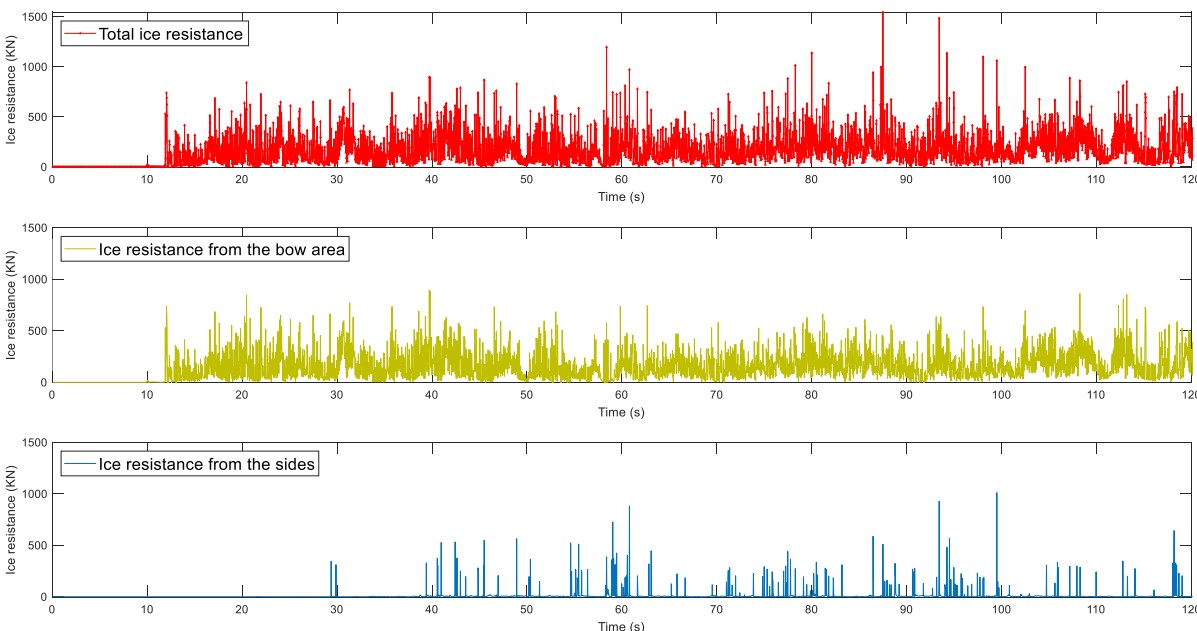

**Figure 9.** Time series of ice resistance when the air-bubble system is off: (**top**) total ice resistance; (middle) ice resistance from the bow area; (**bottom**) ice resistance from the sides.

**Table 3.** Ice resistance of the stable stage.

| Ice Resistance (kN) | Without the Air-Bubble System | With the Air-Bubble System | Drag Reduction Rate |
|---|---|---|---|
| Total | 196.9 | 166.9 | 15.3% |
| Bow area | 180.8 | 162.2 | 10.3% |
| Ship sides | 16.1 | 4.7 | 70.8% |

The effect of the ship speed on the drag reduction rate was also investigated. In addition to the abovementioned ship speed of 6 knots, ship-ice interactions under four other speeds were simulated and ice resistances with and without the air-bubble system were compared. Table 4 listed the resistances as well as the drag reduction rates, which are also plotted in Figure 11. It is observed that the drag reduction rate decreases with the speed increase. This can be explained by the fact that with the increase of the ship's speed, the location where the bubbles reach the free surface moves backwards due to the drag effect of the fluid. This implies the area covered by the gas-water mixture moves backwards at a higher speed, resulting in a larger hull surface in the front in contact with ice. The drag reduction rate is thus reduced. This is however a tentative explanation of this interesting phenomenon. The effect of ship speed on the drag reduction rate of an air-bubble system requires systematic investigation, in a combination of other factors such as the injected air volume, which is included in the authors' future work.

In addition to the drag force in the longitudinal direction, the hull-ice interaction forces in the transverse direction regarding the air-bubble system are also analyzed. Figure 12 shows the time series of ice-induced drift force with and without the air-bubble system. It is seen from the figure that the ice-induced drift forces increase gradually in the first stage of the ice-going voyage for both cases. This is similar to the drag force, which can be explained by the fact that the air-bubble system has not been utilized when the ship enters the ice field. After the entrance stage up to t = 25 s, the ice-induced drift force is found to be smaller in both magnitude and variation. The mean value and the standard deviation of the ice-induced drift force without the air-bubble system are 14.0 kN and 94.3 kN, respectively. For comparison, when the air-bubble system has been activated, the mean and the standard deviation of the ice-induced drift force become 8.4 kN and 64.3 kN, respectively, which

indicates the ice-induced drift force has also been reduced significantly. It is also noticeable that the ice-induced drift force has a large standard deviation. This is because the ice forces on the ship's sides are asymmetric.

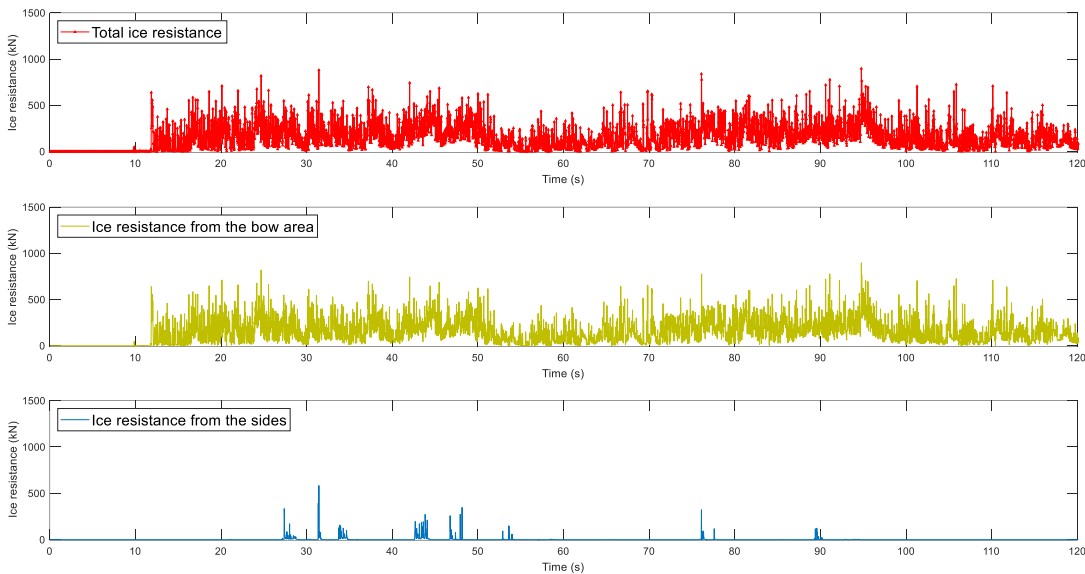

**Figure 10.** Time series of ice resistance when the air-bubble system is on: (**top**) total ice resistance; (**middle**) ice resistance from the bow area; (**bottom**) ice resistance from the sides.

**Table 4.** Ice resistance at different speeds.

| Speed (Knot) | Ice Resistance (kN) (Air-Bubble System off) | Ice Resistance (kN) (Air-Bubble System on) | Drag Reduction Rate |
|---|---|---|---|
| 4 | 121.2 | 96.1 | 20.7% |
| 6 | 196.9 | 166.9 | 15.3% |
| 8 | 291.1 | 257.4 | 11.6% |
| 10 | 372.3 | 341.4 | 8.3% |
| 12 | 500.6 | 480.2 | 4.1% |

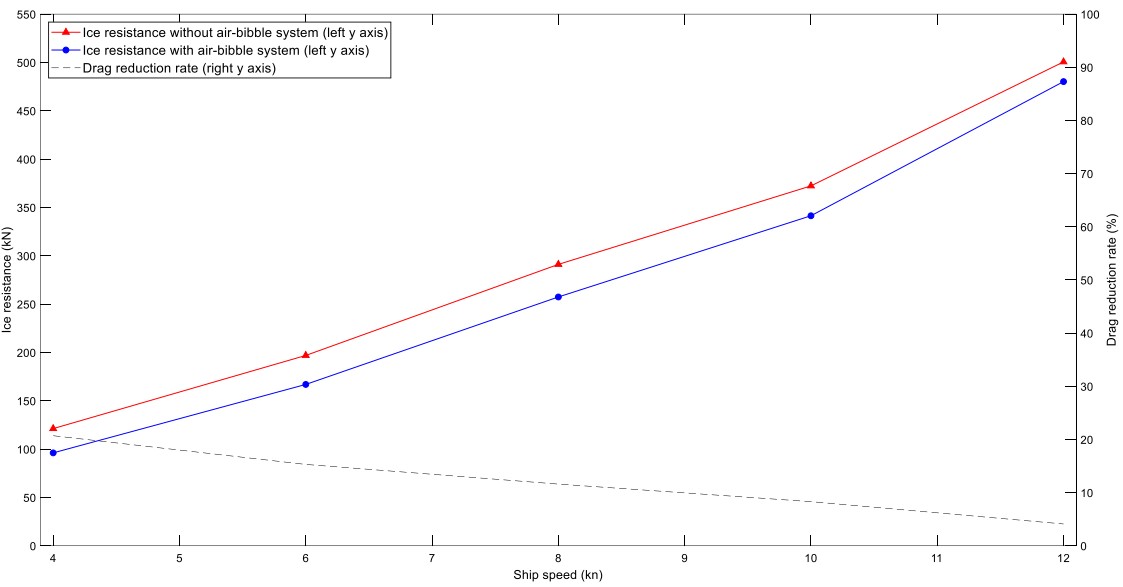

**Figure 11.** The ice resistance and drag reduction rate at different speeds.

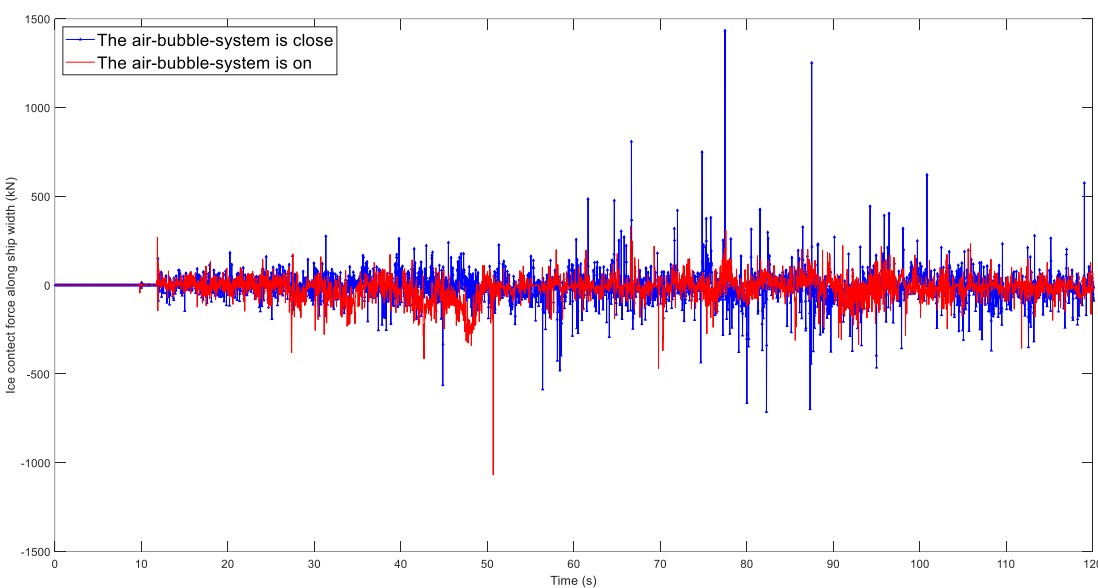

**Figure 12.** Comparison of time history curves of ice contact force along ship width.

### 4.3. Effects of Ventilation Rate

In this sub-section, a sensitivity study about the ventilation rate at the nozzle regarding drag reduction rate was carried out. Different air velocities from the nozzles were investigated under the condition of an ice concentration of 60% and a ship speed of 6 kn. The nozzles are divided into two groups: the bow nozzles are the first 3 pairs of nozzles at the bow area, and the side nozzles are the rest of 5 pairs at the sides that are near the bilge; see Figure 1 for the exact locations of the nozzles. The ventilation rate in m/s represents the gas flow rate at the nozzle. Four air velocities, i.e., 1, 2.5, 5 and 10 m/s, were investigated. A reference case with zero air velocity represents the condition when the air-bubble system is turned off. The drag reduction for a total of eight cases with different air velocities was calculated. The ice resistance data and drag reduction rate under for these cases are listed in Table 5. In this table, Case D represents the air-bubble system with a ventilation rate of 2.5 m/s, which has been mentioned in the previous sub-sections.

**Table 5.** Ice resistance under different air velocities.

| Case | Bow Ventilation Rate (m/s) | Side Ventilation Rate (m/s) | Ice Resistance (kN) | Drag Reduction Rate |
|------|----------------------------|------------------------------|---------------------|---------------------|
| Ref. | 0    | 0    |       | -     |
| A    | 0    | 1    | 187.0 | 5.5%  |
| B    | 1    | 1    | 179.8 | 8.7%  |
| C    | 2.5  | 1    | 170.9 | 13.2% |
| D    | 2.5  | 2.5  | 166.9 | 15.3% |
| E    | 5    | 1    | 152.5 | 22.5% |
| F    | 5    | 5    | 149.5 | 24.1% |
| G    | 10   | 1    | 137.7 | 30.0% |
| H    | 10   | 10   | 132.9 | 32.5% |

It is observed from Table 5, that with the increased ventilation rate the ice resistance is reduced. Let us look closer at the specific cases. Case A is when the bow-nozzles are turned off and the side-nozzle ventilation rate is set as 1 m/s. Figure 13 shows the simulation of the ice resistance time series of this case. When the side nozzles are turned on, air bubbles go up along the hull wall to the free surface. The ice floes at the ship's side become thus overturned and move away from the hull. As a result, the occurrence of hull-ice interaction on both sides of the ship side is reduced. A further step is to turn on the bow nozzles. This is Case B, for which the simulation and ice resistance time series are illustrated in Figure 14.

For this case, the air velocities are set to 1 m/s for all the nozzles. It is seen from Figure 14 that observed the ice-free zone moves forward to the vicinity of the ship's shoulder, and at the same time, the width of the ice-free zone increases. However, the ice accumulation at the bow area remains almost unchanged. Comparing Case B with Case A, the drag reduction effect is not significant.

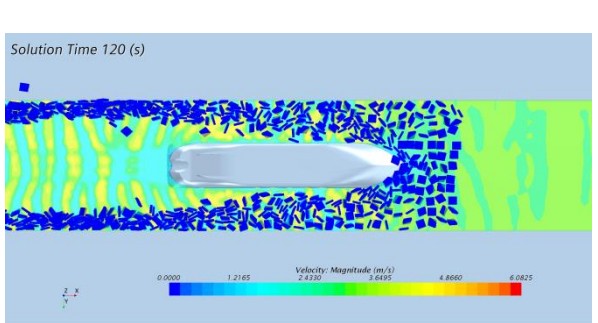
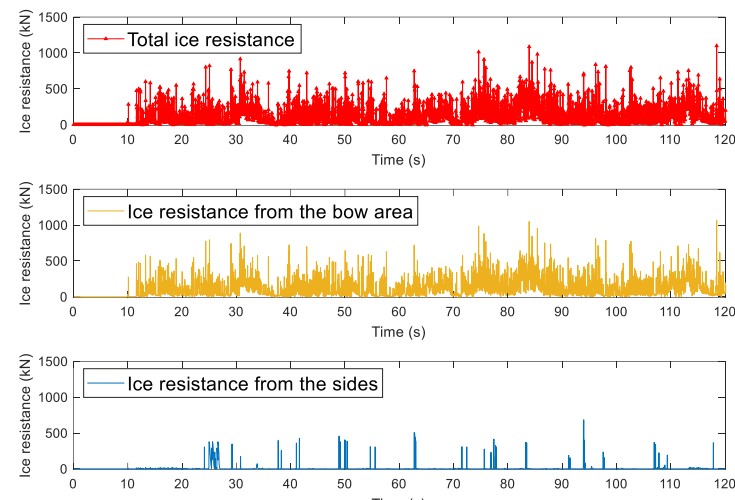

**Figure 13.** Simulation and ice resistance of Case A: the side-nozzle ventilation rate is 1 m/s; the bow-nozzles are turned off.

To better interpret the drag reduction rates in Table 5, the ice resistance values under the various air velocities are paired for comparison, as shown in Figure 15. The cases in Table 5 are put into two categories: one is featured by the side-nozzle ventilation rate kept as 1 m/s; the other is characterized by the nozzles having the same ventilation rate. Comparing the two groups, it is found that the ice resistance of the two groups is quite close under the same bow-nozzle ventilation rate. This implies that the drag reduction effect is more sensitive to the bow ventilation volume. If the bow ventilation volume is sufficiently large, the side ventilation volume has a marginal contribution to drag reduction. This is in line with the fact that for this vessel, the ice resistance on the side of the ship accounts for less than 30% of the total ice resistance.

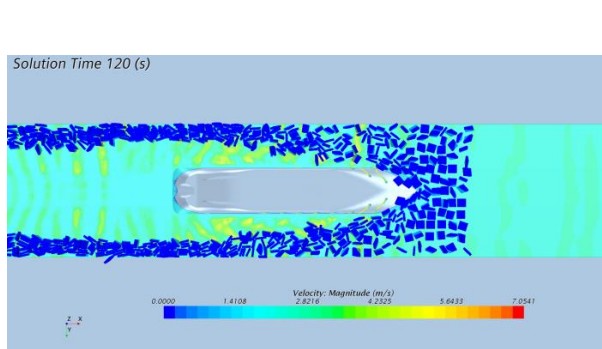
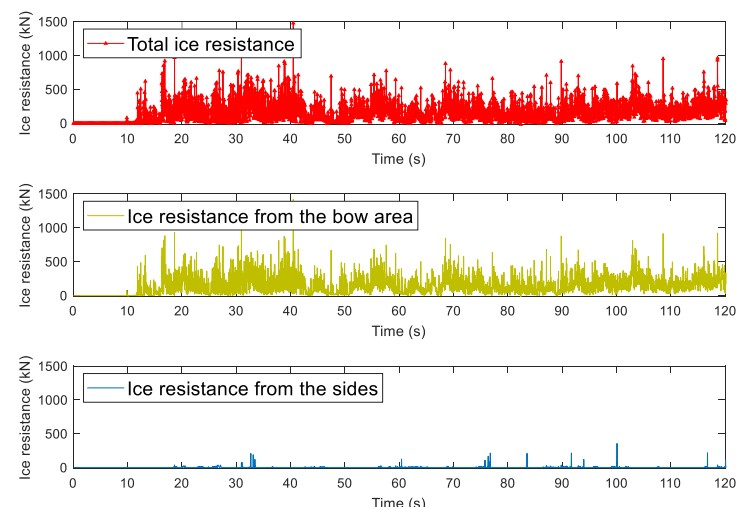

**Figure 14.** Simulation and ice resistance of Case B: the side-nozzle ventilation rate is kept as 1 m/s; the bow nozzles are turned on as 1 m/s.

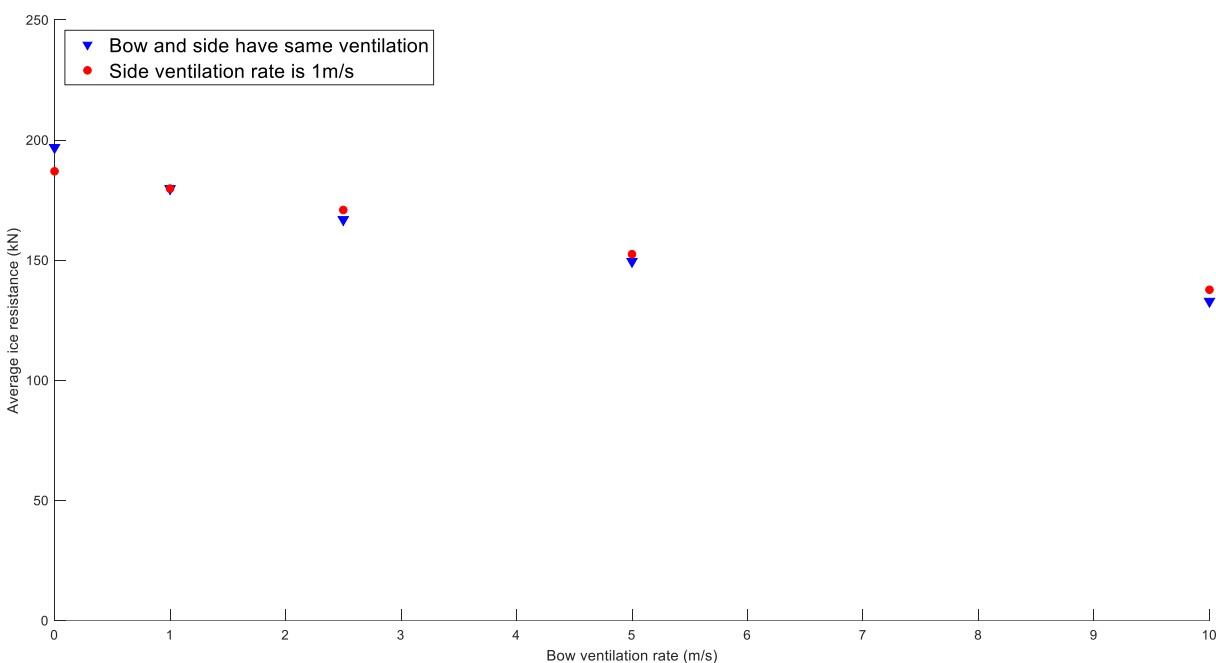

**Figure 15.** Ice resistance comparison for the various air velocities.

The work presented in this article is one of the first investigations on numerical simulation of air-bubble systems regarding drag reduction in floe ice fields. The ice conditions were simplified to ice floes of identical size and shape, which are the delimitations of the current work. An icebreaker was employed as the case study vessel for the demonstration of the proposed procedure. Other ship types with different hull forms need to be modelled to verify the robustness of the proposed procedure. Ice model tests are also required for validation of the numerical results, which are included in the authors' ongoing work. Despite the limitations, the proposed procedure is expected to facilitate design of new generations of ice-going ships.

## 5. Conclusions

In this paper, the authors made use of a coupling CFD-DEM approach in combination with the VOF method to simulate resistance in floe ice fields, aiming to establish a numerical analysis procedure for ice-going ships installed with air-bubble systems. From the simulations and analyses, a more distorted wave making around the hull is observed after turning on the air-bubble system. Ice floes in contact with the hull side wall are pushed away from the hull by the gas-water mixture, resulting in an ice-free zone close to the side hull. The ventilation rate of the air-bubble system is also studied. It is found that the drag reduction rate increases with the increase of ventilation but decreases somewhat at higher speeds. Side ventilation only contributes to reducing the side friction resistance, and the side friction resistance can be eliminated under low ventilation. In general, the bow ventilation plays a deciding role in the overall drag reduction.

**Author Contributions:** Conceptualization, B.-Y.N. and Z.L.; methodology, B.-Y.N. and H.W.; software, Z.L.; validation, H.W.; writing—original draft preparation, H.W.; writing—review and editing, B.-Y.N., Y.X. and Z.L.; project administration, Y.X. and B.F. All authors have read and agreed to the published version of the manuscript.

**Funding:** This research was funded by National Natural Science Foundation of China, grant numbers 52192690, 52192693, 51979051, 51979056, U20A20327; and by National Key Research and Development Program of China, grant number 2021YFC2803400. Computations were partially performed by resources provided by the Swedish National Infrastructure for Computing (SNIC) and were partially funded by the Swedish Research Council through grant agreement No. 2018-05973.

**Institutional Review Board Statement:** Not applicable.

**Informed Consent Statement:** Not applicable.

**Data Availability Statement:** There are no publicly available data for this study.

**Conflicts of Interest:** The authors declare no conflict of interest.

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
