# Peer review of "Numerical Simulation of an Air-Bubble System for Ice Resistance Reduction"

_jmse, doi:10.3390/jmse10091201_

Round 1

Reviewer 1 Report

The authors present a numerical mode coupling CFD and DEM to simulate the effect of the Air-bubble System for Ice Resistance Reduction. Some comments can be considered during the revision and included in the manuscript:

1. The novelty of the work must be clearly presented related to the literature review.

2. There are many works published related to the air-bubble system, and they are not mentioned in the manuscript; I think this will help to have a better overview of what is done in this field.

3. I am wondering why there is no air bubble system at the keel of the ship (along the ship), and it is only installed on the bow and along the side.

4. Are these 6 holes, especially in the bow, enough to produce a suitable amount of air, or do they require more holes?

5. The authors declare that the drag reduction is reduced by around 30%; I am wondering about the power required to force the air (used in the air bubble system) as it will affect the fuel consumption. Is this will be a feasible solution, or in the end, the same higher consumption is required to drive the ship?

6. Due to the intact with ice, is there any trim occurring and is it considered during simulation? 

7. If the speed of the ship increases for instance to 7 or 9 kn, what will be the diff in the results?

8. Conclusion can be rewritten to focus on the main concluded points?

9. The refs can be written according to the journal format.

Author Response

REPLY TO REVIEWERS: JMSE-1863637

We have received feedback from two reviewers on our manuscript. The reviewers have provided us with good feedback with suggestions for improvements and changes to be made in the manuscript. Each issue has been handled separately by the authors and the replies to the feedback are summarized below. Please contact us if the replies are unclear or if the information is missing. Next texts and the texts that have been thoroughly revised are marked with red.

Reviewer 1

The authors present a numerical mode coupling CFD and DEM to simulate the effect of the Air-bubble System for Ice Resistance Reduction. Some comments can be considered during the revision and included in the manuscript:

  1. The novelty of the work must be clearly presented related to the literature review.

Reply: thank you for the positive feedback. As for comment #1, the authors think that the requested information is covered by lines 32 – 92 now. The different paragraphs communicate air lubrication technology in general, an overview of reducing ice resistance by air bubbles, the major numerical methods for simulating air-bubbles, and how different approaches can be utilized to simulate ice-water-hull interactions. The entire introduction part has been revised.

  1. There are many works published related to the air-bubble system, and they are not mentioned in the manuscript; I think this will help to have a better overview of what is done in this field.

Reply: thanks for the suggestion. From the literature study of the authors, the research on the air-bubble system is mainly used to reduce the open water resistance of ships, but much fewer publications on the air-bubble system to reduce ice resistance. And most of the publications about using air-bubble systems to reduce ice resistance are based on experimental approaches. In other words, numerical simulations of ice resistance reduction by an air-bubble system have not been reported.

  1. I am wondering why there is no air bubble system at the keel of the ship (along the ship), and it is only installed on the bow and along the side.

Reply: thanks for a very good question! Conventional air-bubble systems aim at reducing the water resistance of the ship, which requires the air-bubble system to cover the wet surface of the hull as much as possible. For that purpose, the ship bottom and the keel areas need to be covered by air-bubbles. The air-bubble system in this study, in contrast, is supposed to reduce ice resistance instead. It would be sufficient to use a smaller volume of air from the bow/sides to push the crushed ice away from the hull. If we place the holes in the bottom or at the keel, air from the bottom/ keel cannot quickly reach the free surface to generate a gas-water mixture as we expected. This is pointed out in the revised manuscript.

  1. Are these 6 holes, especially in the bow, enough to produce a suitable amount of air, or do they require more holes?

Reply: thanks for the comments! As mentioned previously, the air-bubble system in this study differs from the conventional air-bubble system that reduces water resistance. We in this study aim to reduce ice resistance instead. From the simulation results in Fig. 6(b), the gas-water mixture generated by the bow nozzles is sufficient to produce a good drag reduction effect regarding ice resistance. On the other hand, the authors agree that the effect of the nozzle numbers and size on the reduction of ice resistance requires further research, which is considered in the authors’ planned future work.

  1. The authors declare that the drag reduction is reduced by around 30%; I am wondering about the power required to force the air (used in the air bubble system) as it will affect the fuel consumption. Is this will be a feasible solution, or in the end, the same higher consumption is required to drive the ship?

Reply: thanks for a very good question! In this paper, the 30% drag reduction rate occurs in the case where the jet velocity of the bow and the side of the ship is 10m/s. At this time, the overall jet air volume of the whole ship is about 120000m³/h. According to the reference [12], the actual ship system that adopts the air pump can be roughly converted to the power of the bubble system in this paper as 2100kw, accounting for 14% of the total power (15000kw) of this type of icebreaker. In addition, in practice, it is more likely to take the form of working condition D in this paper (jet velocity 2.5m/s). At this time, the drag reduction efficiency is about 15%, and the power consumption is 525kw, accounting for 3.5% of the total ship power. The authors agree that power consumption is related to ship resistance. However, keeping fuel consumption low is not the only purpose of reducing ice resistance. Sometimes ice-going ships need to sail through heavy ice conditions despite high power and fuel costs. For this reason, we in this work prefer to focus on the drag reduction effect of the air-bubble system and leave other aspects to future research.

  1. Due to the intact with ice, is there any trim occurring and is it considered during simulation? 

Reply: Thanks for the question! We didn’t observe any significant trim for the simulated cases. This however might depend on ship’s bow angle. In our future study, ships of other bow types will be included and this question will be further investigated.

  1. If the speed of the ship increases for instance to 7 or 9 kn, what will be the diff in the results?

Reply: thanks for the valuable comment! The drag reduction effect of the bubble system at different speeds has been increased according to your suggestion. The results are now listed in Table 4 and illustrated in Figure 11. A paragraph (lines 367-380) is also added according to the new results.

  1. Conclusion can be rewritten to focus on the main concluded points?

Reply: Thanks for the comments!  For research like this work, we summarized our findings and trying not to give too strong conclusions.

  1. The refs can be written according to the journal format.

Reply: The format of the reference has been modified according to your suggestions.

Reviewer 2 Report

On page 96, the title of the chapter: "2. The numerical models", should start on page 97 together with the text of the given chapter.

Formula (12) needs to be divided into 2 lines because it is too long.

It is better to describe fig. 4. Describe the numerical throws for red, yellow and green colors. Reduce the width of image 4 or split it into 2 lines.

Adjust the width of fig. 8, 14, 15 or place them under each other.

Why in fig. 16, the values for the speed of 4, 6, 7, 8 and 9 m/s are not indicated.

The article Numerical Simulation of an Air-bubble System for Ice Re-2 Sistance Reduction is of high quality. It shows numerical results using simulation. the article brings new things in the scientific field and after minor graphic and formal adjustments I recommend the article for publication.

In the article, I highlight the literature used. It is of high quality for the given theme.

Author Response

REPLY TO REVIEWERS: JMSE-1863637

We have received feedback from two reviewers on our manuscript. The reviewers have provided us with good feedback with suggestions for improvements and changes to be made in the manuscript. Each issue has been handled separately by the authors and the replies to the feedback are summarized below. Please contact us if the replies are unclear or if the information is missing. New texts and the texts that have been thoroughly revised are marked in red.

Reviewer 2

  • On page 96, the title of the chapter: "2. The numerical models", should start on page 97 together with the text of the given chapter.

Reply: thanks for the comment. It has been modified according to your suggestions.

  • Formula (12) needs to be divided into 2 lines because it is too long.

Reply: thanks for the comment! Formula (12) has been changed to two lines as you suggested.

  • It is better to describe fig. 4. Describe the numerical throws for red, yellow and green colors. Reduce the width of image 4 or split it into 2 lines.

Reply: thanks for the comment! The description of the meaning of the colors has been added and the image width has been reduced.

  • Adjust the width of fig. 8, 14, 15 or place them under each other.

Reply: thanks for the comment! The images have been resized as you suggested to make sure the individual images are below each other.

  • Why in fig. 16, the values for the speed of 4, 6, 7, 8 and 9 m/s are not indicated.

Reply: thanks for the question. Because the main research content of this paper is the influence of the airflow velocity of the bow air holes and the side air holes on the ability to reduce ice resistance, it is hoped that there will be a set of extreme working conditions to reflect the difference in importance between the bow air holes and the ship side air holes. If the extreme working condition group and other working conditions are traversed, the calculation time of the numerical simulation will be greatly increased, so this paper selects additional working conditions in Table 4 now, and does not consider the cases of 4, 6, 7, 8, and 9 m/s in Fig.16 (now Fig.15) further.

The article Numerical Simulation of an Air-bubble System for Ice Resistance Reduction is of high quality. It shows numerical results using simulation. the article brings new things in the scientific field and after minor graphic and formal adjustments I recommend the article for publication.

In the article, I highlight the literature used. It is of high quality for the given theme.

Reply: thank you for the positive feedback very much.

Round 2

Reviewer 1 Report

The authors replied to all comments and I recommend the paper to be published.